# Building a better understanding of labour exploitation's impact on migrant health: An operational framework

Sabah Boufkhed[1,2]*, Nicki Thorogood[3], Cono Ariti[4,5], Mary Alison Durand[1]

**1** Department of Health Services Research and Policy, Faculty of Public Health and Policy, London School of Hygiene & Tropical Medicine, London, United Kingdom, **2** Humanitarian and Conflict Response Institute, The University of Manchester, Manchester, United Kingdom, **3** Department of Public Health, Environments and Society, Faculty of Public Health and Policy, London School of Hygiene & Tropical Medicine, London, United Kingdom, **4** Department of Medical Statistics, Faculty of Epidemiology and Population Health, London School of Hygiene & Tropical Medicine, London, United Kingdom, **5** Centre for Medical Education, Cardiff University School of Medicine, Cardiff, United Kingdom

* sabah.boufkhed@manchester.ac.uk

## Abstract

### Background

There is limited evidence on labour exploitation's impact on migrant health. This population is, however, often employed in manual low-skilled jobs known for poor labour conditions and exploitation risks. The lack of a common conceptualisation of labour exploitation in health research impedes the development of research measuring its effects on migrant health and, ultimately, our understanding of migrants' health needs.

### Aim

To develop an operational conceptual framework of labour exploitation focusing on migrant workers in manual low-skilled jobs.

### Methods

Non-probabilistic sampling was used to recruit multidisciplinary experts on labour exploitation. An online Group Concept Mapping (GCM) was conducted. Experts: 1) generated statements describing the concept 'labour exploitation' focusing on migrants working in manual low-skilled jobs; 2) sorted generated statements into groups reflecting common themes; and 3) rated them according to their importance in characterising a situation as migrant labour exploitation. Multidimensional Scaling and Cluster Analysis were used to produce an operational framework detailing the concept content (dimensions, statements, and corresponding averaged rating).

### Findings

Thirty-two experts sorted and rated 96 statements according to their relative importance (1 "relatively unimportant" to 5 "extremely important"). The operational framework consists of four key dimensions of migrant labour exploitation, distributed along a continuum of severity

**Data Availability Statement:** All relevant data are within the paper and its supporting information.

**Funding:** This research is part of Sabah Boufkhed's doctoral research, which received funding from the

UK Economic and Social Research Council (ref. 1355496) and complementary funds from the Fund for Women Graduates (ref. 17037). The funders had no role in study design, data collection and analysis, decision to publish, or preparation of the manuscript.

**Competing interests:** The authors have declared that no competing interests exist.

revealed by the rating: 'Shelter and personal security' (rating: 4.47); 'Finance and migration' (4.15); 'Health and safety' (3.96); and 'Social and legal protection' (3.71).

## Conclusion

This study is the first to both generate an empirical operational framework of migrant labour exploitation, and demonstrate the existence of a "continuum from decent work to forced labour". The framework content can be operationalised to measure labour exploitation. It paves the way to better understand how different levels of exploitation affect migrant workers' health for global policymakers, health researchers, and professionals working in the field of migrant exploitation.

## Introduction

Manual low-skilled jobs, in which migrant workers are mostly employed, present high occupational health hazards (e.g. chemicals or accidents), and are often referred to as 'exploitative' [1–5]. Migrant workers in these jobs are at high risk of being severely exploited [6–8], and may face serious health consequences [8–10].

Reviews of the evidence on migration and health reveal important gaps in the literature on migrant workers' health [11–13]. Yet, they represent almost two-thirds of international migrants and the International Labour Organization (ILO) estimated that there are 150 million migrant workers worldwide [14]. The ILO defines migrant workers, as "international migrants who are currently employed or are unemployed and seeking employment in their present country of residence" [8, p.xi]. This paper focuses on international migrant workers.

Global political agendas and the health sector have increasingly demonstrated an increased interest in issues of migrant health and labour exploitation [15–17]. In 2019, the World Health Organisation (WHO) called for the development of policies and interventions and for better monitoring systems to improve migrant health [17]. The United Nations (UN) set Sustainable Development Goals (SDGs) for countries to reach by 2030. They include promoting 'health for all' (SDG 3), decent work (SDG 8), and safe and fair migration (SDG 10), along with fighting against migrant workers' exploitation and modern slavery [18]. A rise in research on migrant workers' exploitation can therefore be anticipated in the coming years.

However, a conceptual framework of migrant labour exploitation that would clarify the concept and be operationalisable for research and action is still missing. Such a framework is essential to support health research, and inform policies and interventions intended to improve migrant health. This study formed part of the first author's doctoral thesis, which [19] aimed to develop an operational conceptual framework of labour exploitation focusing on migrant workers in manual low-skilled jobs, building on a continuum approach. The study presented here aimed to clarify the concept content by identifying the dimensions; subdimensions and statements constituting migrant labour exploitation as viewed by professional experts working in the field. We argue that the continuum approach can overcome epistemological divergences in the field that will now be described.

### The concept of labour exploitation and the exploitation continuum

The term 'labour exploitation' has been used to describe situations ranging from indecent, harsh or unfair labour conditions, to those of modern slavery, human trafficking or forced

labour [20–23]. Key characteristics of labour exploitation may include underpayment, poor working conditions and safety, and violations of labour rights [21]. For some, exploitation implies the use of coercion [24], a person unfairly benefiting from another [25], or harm to the victim [26]. The global 'exploitative' working conditions of migrant workers, which have been mainstreamed by international non-governmental organisations (NGOs), media and international civil society, mostly refer to human rights violations [6, 22, 27–31]. Furthermore, the literature on working and living conditions of migrant workers in low-skilled jobs implicitly refers to labour exploitation without precisely defining what *constitutes* labour exploitation [32–34]. Exploitation seems to portray migrant workers' poor conditions, wages issues, high workload and how they are poorly treated, without clear specifications.

Skrivankova (2010) suggests that situations of labour exploitation can be conceptualised along a "continuum between decent work and forced labour" (p.1). She claims that it facilitates the understanding of such a complex social concept while acknowledging the range of experiences of exploitation [22]. In her theoretical conceptualisation of exploitation for the legal field, situations located closer to breaches of decent work standards could be prosecuted under labour law. We will refer to this as the 'lower part' of the continuum. At the other end of the continuum, criminal and human rights laws could be used to address the most severe situations of exploitation, such as forced labour, human trafficking, or modern slavery. We will refer to this end as the 'extreme part'.

Our application of this continuum-based conceptualisation revealed a conceptual gap in the public health literature. It reveals that, to date, two main schools of thought in public health have discussed issues of labour exploitation: 1) the Human Rights school of thought, focusing on the extreme part of the continuum (close to forced labour); and the Social Determinants of Health school of thought, focusing on the lower part (close to decent work breaches). The name we use to describe these schools of thoughts (Human Rights and Social Determinants of Health schools) reflects the approaches they take to address labour exploitation.

## Two schools of thoughts in health research on labour exploitation

The Human Rights (HR) school of thought incorporates research on issues of labour exploitation related to human trafficking and modern slavery. It covers public health literature on extreme forms of labour exploitation [6, 11], and focuses on individuals' risks and exposures to hazards. The HR school has focused on migrant workers, and mainstreamed and positioned extreme forms of labour exploitation as a public health issue [35, 36]. In 2017, PLoS Medicine released a special issue on '*Human Trafficking, Exploitation and Health*', which framed these issues as human rights violations [37]. Studies within this school have described the extreme labour conditions faced by—mostly migrant—workers, such as exposure to physical and psychological violence, and coercion or restriction of freedom, along with the serious health issues they face [8, 9, 38]. To study this issue, the HR school uses a variety of indicators of (extreme) exploitation [9, 39], such as the ILO's Indicators for Forced Labour which focus on coercion and restriction of freedom [40]. Within this school, terms like human trafficking, forced labour or slavery are used interchangeably to describe criminal forms of exploitation [41–44]. Each of these terms, however, refers to different and specific legal frameworks, such as the Palermo protocol, the 1930 ILO convention against Forced Labour and the 1926 Slavery convention [45–49], which outlaw specific forms of labour exploitation, mostly those where perpetrators use coercion, deception or restriction of freedom of movement [50–53]. Each framework defines a 'victim' status, and hence conditions his/her health and social rights.

The Social Determinants of Health (SDH) school, on the other hand, has focused on the role of structures or social determinants, like employment and working conditions, in creating

or enabling labour exploitation, which in turn may determine workers' health. Social epidemiologists participating in the Employment Conditions Network (EMCONET) [54], for example, are part of this school. They have used a Marxist approach and conceptualised exploitation as a social mechanism [54, 55]. Labour exploitation has been operationalised as an organisational factor [56] or a relational determinant of health [57]. Muntaner et al. [56], for example, use *"a firm's owner-ship type (e.g., for-profit vs. not-for-profit/nonprofit) [...] as an organizational-level indicator of social class exploitation"* [56; p.28]. Contrasting with the HR school, this school regards slavery and trafficking as non-standard forms of employment, like precarious work [54, 58]. The literature on precarious employment often refers to workers' exploitation [59, 60], and suggests that this form of employment should be considered when studying exploitation. For example, Muntaner [61] describes features of precarious employment which echo mainstream views on 'labour exploitation': *"uncertainty in the duration of the labor contract, [...] psychological job insecurity, employment strain, low wages and lack of benefits, hazardous physical and psychological working conditions, and de facto or real absence of legal protection"* [61; p.2].

## Continuum and social epidemiological approach

Berkman et al. [62] suggest that "*population distributions for most [social] risk factors move along a continuum with a normal distribution*" [53, p.7]. Exposure to labour exploitation among migrant workers in manual low-skilled jobs is, therefore, more probably distributed along a continuum instead of being binary (i.e. exploited or not). Further, migrants working in sectors 'known to be exploitative' who are identified as 'trafficked' may have similar health needs to those who are not or not [8]. In fact, researchers within the HR school, who have focused on migrant workers, have begun to take a continuum approach when studying extreme forms of labour exploitation [37, 63]. They have increasingly referred to structural and occupational aspects of exploitation, traditionally the focus of the SDH school.

The case of migrant workers underscores the need for a continuum approach in health research to overcome issues related to a binary approach. Chapkis [64] argued that the roots of the USA's fight against human trafficking are founded in anti-immigration discourses and moral values. The mainstream anti-trafficking fight using criminal justice, she argues, distinguishes two groups of migrants: those identified as 'trafficked' and therefore deserving support on social justice grounds, and the 'less-deserving' and largest group migrating for economic purposes. A continuum approach could help us centre on exploited workers' needs and overcome the issue that governments can potentially view them either as crime victims (e.g. trafficked) or as crime perpetrators, should they have an irregular immigration status [64, 65].

To address "complex social and health problems" like exploitation and design social epidemiological studies aimed at informing future actions (e.g., interventions or policies), Cwikel [66] recommends the SOCEPID framework. It describes three key research phases: 1) development of a conceptual framework; 2) conducting research and data collection; and 3) applied social epidemiology. The larger research project (SB's doctoral thesis [19]) within which this study is nested addresses the first phase: to develop a culturally sensitive conceptual framework, which would inform future epidemiological research on the health impacts of labour exploitation on migrant workers (phases 2 and 3). In addition to the framework described in this paper, another conceptual framework was developed from the perspective of Latin Americans working in manual low-skilled jobs in London to explore potential cultural and contextual specificities of labour exploitation [19]. The current paper focuses on the experts' framework, which is intended to generate a standardisable measurement framework of 'labour exploitation' that can be adapted to different contexts and populations, starting with (international) migrant workers in low skilled jobs.

## Methodology and concept mapping method

We drew on a mixed-methods methodology [67] to address the complexity of defining the content and dimensions of labour exploitation as a concept which involves different approaches to exploitation, methods and stakeholders. Group Concept Mapping (GCM) is an intrinsically participatory mixed-method combining qualitative data collection (brainstorming and sorting-rating exercise to generate the concept content) with multivariate analyses (multi-dimensional scaling and cluster analysis to organise the content into dimensions) [68]. It consists of six phases: (1) preparation, (2) brainstorming, (3) sorting-rating exercise, (4) multivariate analysis, (5) interpretation, and (6) utilisation phase (see Fig 1). It implies the collection and combination of *various* stakeholders' inputs to produce an operationalisable (structured) conceptual framework, which clarifies the content of abstract concepts [69, 70]. GCM, therefore, differs from other concept mapping methods that visually map knowledge and concepts based on *one* individual's thoughts [71]. Further, GCM is used to design measures [72]. The methodology is detailed in full in SB's doctoral thesis [19].

---

**1 - Preparation**
- Sampling, recruitment, definition of the focus questions and prompts for brainstorming, sorting and rating

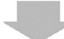

**2 - Generation of statements**        (data collection 1)
- Brainstorming
    Experts produce statements describing the concept content
- *The Researcher performs data reduction and synthesis to reduce the list of statements for next phase*

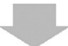

**3 - Structuring of ideas**        (data collection 2)
- Sorting-rating exercise
    Participants group the statements "in a way that makes sense to them"
    Participants rate each statement according to its importance in defining a situation as labour exploitation

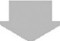

**4 - Statistical analysis**
- Multidimensional scaling (MDS)
    Production of a point-map on which the points (statements) close to each other are conceptually similar
- Cluster analysis (CA)
    Identification of the concept dimensions by regrouping nearby points (conceptually similar statements) into clusters

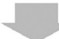

**5 - Interpretation & 6 - Utilisation**
- Labelling the clusters on the maps
- Identifying regions of meaning
- Identifying the use of the maps

---

**Fig 1. Group concept mapping steps adapted for the research.** Source: Adapted from Trochim W. An Introduction to Concept Mapping for Planning and Evaluation. Eval Program Plann. 1989 Jan 1;12(1), p. 3.

The GCM we conducted in this study resulted in what we have termed the expert 'skeleton' framework of migrant labour exploitation. The notion of *"expert skeleton"* was coined by Novak and Cañas [73] for their individual concept mapping method in the field of education. Our *"expert skeleton"* map used multidisciplinary expertise to describe the key content of the concept of labour exploitation focusing on migrant workers in manual low-skilled jobs. We see it as a standardisable framework that could be adapted or expanded for different contexts and populations, to comprehensively understand labour exploitation. It can be elaborated by integrating migrant workers' voices through further GCM, thereby leading to the delineation and addition of new dimensions or items which encapsulate their knowledge and lived experiences. The latter will be the focus of another paper.

## Population and sampling

In the current study, the term 'expert' refers to academic and non-academic professionals (e.g. in international organisations, NGOs, workers' organisations) in the field of labour exploitation, and meeting at least one of the following criteria: participated in the development of a measure related to labour exploitation; or worked for at least five years within the field; or developed a widely used conceptual framework or operational definition of concepts related to labour exploitation (e.g. forced labour); or referred to as an expert in the literature or mass media. Migrant workers are also considered experts on the topic [74], and we have undertaken further research with international migrants working in low skilled jobs in London to assess their conceptualisation [19].

Purposive sampling was used, with the intention of recruiting at least 16 experts covering both academic and non-academic expertise, both the lower (e.g. precarious work) and the extreme part of the continuum (e.g. modern slavery), and at least one expert per the following field category and disciplines: 1) health; 2) sociology, social sciences, or social work; 3) policy, law or advocacy; 4) economy, finance or business. GCM developers suggest aiming for 10 to 40 participants to "achieve a broad sampling of ideas rather than a representative sampling of persons" [68; p.36]. A second recruitment phase was conducted before the sorting-rating phase as we anticipated that some participants in the first phase might drop out. It is expected that the statements generated for the sorting-rating exercise are varied enough to cover all aspects of the concept to be mapped, and it is common to include new participants who did not participate in the first phase [75, 76]. Identified experts were invited to participate by email, provided with written information about the study, a consent form and the opportunity to speak to and ask questions about the study to the first author (SB).

## Ethical statement

Ethical approval was obtained from the London School of Hygiene and Tropical Medicine's Ethics committee (reference 8698). Written informed consent was obtained before data collection.

## Data collection and management

The main data collection tool used was an online platform that was developed specifically for the brainstorming and sorting-rating phases. The platform was composed of two interfaces: one for participants to enter data, and one for the first author (SB) to manage the content of the platform and data generated. Each data collection phase was piloted. However, some experts reported issues while trying to perform the sorting-rating tasks online. As an alternative, they conducted these steps on an Excel file with the same content as the online platform.

For the first step, participants accessed a demographic page on the platform, and the brainstorming page where they were instructed to generate as many short statements as they wanted to describe the concept, by completing the prompt *"A migrant worker in manual-low-skilled jobs is exploited when..."*. They were instructed that each statement generated should contain one idea. After the task was completed, statements were downloaded and, as they often consisted of more than one idea, they were processed by the first author (SB) so that one statement represented only one idea (extension phase). Data reduction and synthesis of these extended statements were then conducted to remove duplicates and statements which were deemed too vague or out of the scope. The final list of statements was discussed by the four authors until a list of statements (one idea per statement) covering all the unique ideas generated was obtained.

For the second step, participants were asked to structure these statements online by individually participating in the sorting-rating exercise. They first sorted all the statements into groups, *"in a way that makes sense for [them]"* [77], using the following GCM rules [68]: all statements must be sorted; all statements cannot be put into one single group; a group needs to contain at least two statements; one statement can only be placed in one group; there cannot be one group containing only items that would not fit in other groups created ("miscellaneous" group). Participants were advised to label the groups of statements to illustrate the underlying common theme of each group they created. Second, they rated all the statements in terms of their relative importance in characterising a situation as exploitation of migrant workers (Likert scale ranging from 1 "Relatively unimportant" to 5 "Extremely important"). Participants' contributions were exported as Excel files (.csv), and which were then imported into Stata version 13 (Stata Corporation [www.stata.com]) where the data were merged and cleaned.

## Data analysis

Multivariate analyses were performed on the sorting outcomes (groups of statements generated) using SPSS version 24 (IBM SPSS software [www.spss.com]) to generate the concept maps.

First, non-metric multidimensional scaling (MDS) was used to analyse the sorting outcomes. MDS is an iterative data reduction process that transforms a matrix of (dis-)similarities into a two-dimensional map [78, 79]. The latter is the GCM "point map" [77]. Outputs from the sorting exercise (i.e. individuals' groups of statements) were transformed using R (version 14) as a similarity matrix summarising how many times statements were sorted together. The MDS outcome was a list of coordinates for each statement (point). On the point-map generated, the shorter the distance between points, the more conceptually similar the statements. The final MDS model was validated by comparing its stress value to other existing published GCM and by comparing its characteristics to other GCM studies [76, 80]. The stress value *"reflect[ing] the degree to which the conceptualized model (i.e., the concept map) reflects the judgments made by participants as a function of the sorting procedure"* [75; p. 270–71] was assessed. A lower stress value indicates a better model.

Second, hierarchical agglomerative cluster analysis (CA) using Ward's algorithm was performed on the MDS coordinates of each statement (point) to regroup them into conceptually similar clusters [69, 81]. This generated "cluster point maps" [77]. The final cluster point map was selected when the statements composing the clusters shared a common theme (i.e. they were conceptually similar). Once the final solution was identified, the clusters were named using some of the labels provided by participants. The cluster content was refined by checking whether some statements located at the edge of a cluster would better match a nearby cluster, in which case the statement was relocated [69].

The ratings for each statement were averaged and used to weight each statement on the map. Statement ratings were then averaged within each cluster, to generate a cluster-rating map, indicating the importance of each cluster for experts. On the weighted cluster point map, adjacent clusters sharing an underlying meaning were identified and uncovered "regions of meaning" [69], which correspond to key dimensions of the concept. The resulting map was the operational conceptual framework, which detailed the content of labour exploitation: dimensions corresponding to regions of meaning, subdimensions corresponding to the clusters identified in the CA, and the statements (points) generated by the experts during the brainstorming (as per the final list).

## Results

### Participants

A total of 180 experts identified met the selection criteria. Nineteen percent (n = 34/180) consented to participate, and 32 experts participated in at least one phase of the data collection. Sixty-six percent (n = 21/32) participated in both brainstorming and sorting-rating phases. Participants' characteristics are presented in Table 1. Their expertise covered both the lower and extreme parts of the continuum of labour exploitation, and all expected disciplines. Half were academic and half non-academic professionals. At least one participant per continent was included, with half of the sample based in the UK.

### The operational framework and the concept content

Fig 2 depicts the operational 'expert skeleton' framework of labour exploitation focusing on migrant workers in manual low-skilled jobs. It reveals four key dimensions of labour exploitation: 'Shelter and personal security'; 'Finance and migration'; 'Health and safety'; and 'Social and legal protection'. The stress value for the model was 0.18, which compared favourably to those in previous GCM studies [76, 80].

In Fig 2, points represent the statements, blue regions the key dimensions, and unshaded areas the subdimensions. The figure also displays the cluster ratings. The corresponding statement numbers are presented in Table 2 detailing the key dimensions, subdimensions and statements, along with their importance ratings. Clusters with the highest importance to characterise a situation as exploitation are located towards the left of the map, and those with the lowest towards the right. The dimensions and subdimensions will be presented from higher to lower importance rating.

**'Shelter and personal security'.** This dimension is the highest rated and comprises four subdimensions. The *'Physical and psychological intimidation'* subdimension describes severe or harmful situations and mistreatment at several levels: physical, psychological, and financial. *'Deprived of basic needs'* includes statements describing a lack of provision of workers' basic needs, such as being provided with "appropriate food and water", or "being treated cruelly". *'Restriction of freedom and movement'* refers mostly to situations of coercion and indicates some level of restriction of freedom and movement (e.g. *"s/he has his/her identity documents withheld"*). *'Dependence on the job'* comprises statements indicating social isolation and dependence on the job, as well as statements describing the illegal nature of jobs.

**'Finance and migration'.** This dimension is composed of three subdimensions including situations that seem specific to migrant workers. The *'Deductions and migrant work'* subdimension contains statements describing situations of salary withholding, deductions that seem unfair, as well as situations specific to immigration status. *'Wage issues'* covers situations such as being unpaid, underpaid or irregularly paid. *'Misled'* describes situations of workers being misled or deceived about their conditions or rights.

**Table 1. Participants' characteristics.**

| | Overall (N = 32) | | Brainstorming (N = 28) | | Sorting-rating (N = 25) | | Both (N = 21) | |
|---|---|---|---|---|---|---|---|---|
| | n | % | n | % | N | % | n | % |
| **Academics** [1] | 16 | 50.0 | 15 | 53.6 | 12 | 48.0 | 11 | 52.4 |
| **Main discipline or domain of expertise** | | | | | | | | |
| Health | 7 | 21.9 | 6 | 21.4 | 6 | 24.0 | 5 | 23.8 |
| Sociology, social sciences or social work | 7 | 21.9 | 6 | 21.4 | 6 | 24.0 | 5 | 23.8 |
| Economy, finance or business | 1 | 3.1 | 1 | 3.6 | 1 | 4.0 | 1 | 4.8 |
| Policy, law or advocacy | 11 | 34.4 | 9 | 32.1 | 7 | 28.0 | 5 | 23.8 |
| Other | 6 | 18.8 | 6 | 21.4 | 5 | 20.0 | 5 | 23.8 |
| **Part of the hypothesised continuum of 'labour exploitation' covered** | | | | | | | | |
| Lower part [2] | 10 | 31.3 | 9 | 32.1 | 8 | 32.0 | 7 | 33.3 |
| Severe part [3] | 16 | 50.0 | 14 | 50.0 | 11 | 44.0 | 9 | 42.9 |
| Mixed [4] | 5 | 15.6 | 5 | 17.9 | 5 | 20.0 | 5 | 23.8 |
| Missing | 1 | 3.1 | - | - | 1 | 4.0 | - | - |
| **Female** | 17 | 53.1 | 14 | 50.0 | 14 | 56.0 | 11 | 52.4 |
| **Countries** | | | | | | | | |
| Argentina | 1 | 3.1 | 1 | 3.6 | 1 | 4.0 | 1 | 4.8 |
| Australia | 2 | 6.3 | 0 | 0.0 | 2 | 8.0 | 0 | 0.0 |
| Austria | 2 | 6.3 | 2 | 7.1 | 2 | 8.0 | 1 | 4.8 |
| Belgium | 1 | 3.1 | 1 | 3.6 | 0 | 0.0 | 0 | 0.0 |
| Brazil | 1 | 3.1 | 1 | 3.6 | 1 | 4.0 | 1 | 4.8 |
| Costa Rica | 1 | 3.1 | 1 | 3.6 | 1 | 4.0 | 1 | 4. 8 |
| France | 1 | 3.1 | 1 | 3.6 | 1 | 4.0 | 1 | 4. 8 |
| Nepal | 2 | 6.3 | 2 | 7.1 | 2 | 8.0 | 2 | 9.5 |
| Nicaragua | 1 | 3.1 | 0 | 0.0 | 1 | 4.0 | 0 | 0.0 |
| Senegal | 1 | 3.1 | 1 | 3.6 | 1 | 4.0 | 1 | 4. 8 |
| Spain | 1 | 3.1 | 1 | 3.6 | 1 | 4.0 | 1 | 4. 8 |
| UK | 17 | 53.1 | 16 | 57.1 | 11 | 44.0 | 10 | 47.6 |
| USA | 1 | 3.1 | 1 | 3.6 | 1 | 4.0 | 1 | 4. 8 |

For the purpose of this study

[1] defined as researchers who are part of a University

[2] includes precarious, low-paid, insecure, migrant work

[3] includes human trafficking, slavery, modern slavery, forced labour

[4] defined as lower and severe exploitation

**'Health and safety'.** This is a single dimension that covers health, safety and psychosocial hazards. Statements included exposure to unhealthy and unsafe working environments and not being provided with protective equipment or training. It also covers psychosocial hazards, such as *"s/he can be harassed"*, or *"his/her work contract is not renewed unless s/he works extra hours unpaid"*, where workers have to face uncertain futures or accept situations they would normally not because they may fear losing their jobs.

**'Social and legal protection'.** This dimension is the lowest rated and contains five subdimensions. The *'Time-off and legality issues'* subdimension covers a lack of days off in general or of specific time-off for sick or care leave. This cluster also includes statements referring to employment laws violations. *'Contract and workload'* describes issues related to lack of contract or poor contractual arrangements, as well as statements depicting intense working days.

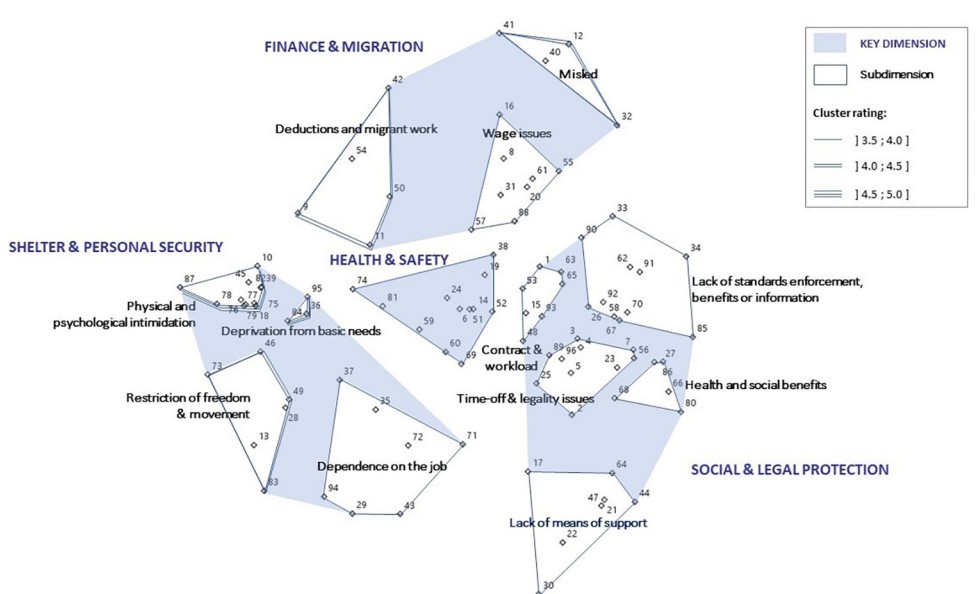

**Fig 2. Operational framework of labour exploitation, focusing on migrant workers in manual low-skilled jobs.**

One subdimension is specific to poor *'Health and social benefits'*, and the *'Lack of means of support'* subdimension contains statements that seem related to causes or facilitators of labour exploitation. They cover a lack of ways to get support or to complain about employment or working conditions, and a lack of access to organisations able to provide help to workers facing issues. *'Lack of standards enforcement, benefits or information'* relates to not benefiting from what generally would be expected for decent or basic employment conditions. It includes statements related to breaches of standards as well as workers' lack of information about their rights.

## A gradient of severity: Statements and ratings

Table 2 reveals a gradient of severity in the distribution of individual statement ratings, where highly rated statements represent particularly severe and harmful situations of exploitation, and the lowest ratings relatively milder forms of labour exploitation related to employment conditions. Higher rated statements have smaller standard deviations than the lower rated, suggesting quite a high agreement about the importance of the highly-rated statements and more disagreement about the lower-rated statements in terms of defining situations of labour exploitation. The values of the average ratings were close to each other and there is no large gap in the ratings given. This suggests a continuous increase in the importance of items composing the concept.

## Discussion

To our knowledge, our study offers the first operational framework of labour exploitation focusing on migrant workers in manual low-skilled jobs. It details the concept content for health research by revealing four main dimensions and detailing their components. It provides a common understanding of labour exploitation that could inform research and policy, and addresses the conceptual and operational gaps that have resulted in a lack of standardisation in health research on labour exploitation.

**Table 2. Content of the concept of labour exploitation focusing on migrant workers in manual low-skilled jobs.**

| ID | KEY DIMENSIONS | Rating | |
|---|---|---|---|
| | Subdimensions and statement labels | Mean | SD |
| | **SHELTER AND PERSONAL SECURITY** | **4.47** | |
| | **Physical and psychological intimidation** | **4.77** | **0.16** |
| 10 | s/he faces criminal levels of abuse | 5.00 | 0.00 |
| 39 | s/he is coerced into continuing to work through debt | 4.88 | 0.33 |
| 78 | s/he is coerced to remain in working conditions that are financially harmful | 4.88 | 0.33 |
| 75 | s/he experiences physical abuse | 4.88 | 0.61 |
| 77 | s/he is coerced to remain in working conditions that are physically harmful | 4.80 | 0.50 |
| 76 | s/he experiences sexual abuse | 4.79 | 0.72 |
| 79 | s/he is coerced to remain in working conditions that are psychologically harmful | 4.72 | 0.54 |
| 82 | s/he works under threat of punishment | 4.72 | 0.54 |
| 87 | s/he is threatened with deportation | 4.72 | 0.68 |
| 18 | s/he experiences violence in the workplace | 4.71 | 0.86 |
| 45 | s/he is in a situation where s/he is exposed to threats | 4.36 | 0.86 |
| | **Deprived of basic needs** | **4.70** | **0.31** |
| 95 | s/he is obliged to work under cruel or inhumane conditions | 4.92 | 0.28 |
| 84 | s/he is forced to work without appropriate access to food and water | 4.84 | 0.37 |
| 36 | s/he is living in the same place as s/he works with inadequate food | 4.35 | 0.93 |
| | **Restriction of freedom and movement** | **4.47** | **0.39** |
| 46 | s/he has his/her identity documents withheld | 4.84 | 0.37 |
| 73 | s/he is obliged to live in cruel, inhumane or degrading conditions | 4.84 | 0.47 |
| 49 | his/her communication outside working hours are curtailed | 4.64 | 0.49 |
| 28 | his/her contact with family is restricted | 4.48 | 0.82 |
| 13 | s/he is vulnerable because of criminal activity involved | 4.04 | 1.34 |
| 83 | s/he is unable to leave because of geographic isolation | 3.96 | 1.24 |
| | **Dependence on the job** | **3.95** | **0.35** |
| 35 | s/he is living in the same place as s/he works with no access to a bathroom | 4.44 | 0.71 |
| 71 | s/he is deprived of access to health services | 4.36 | 0.86 |
| 94 | s/he is dependent on the employer | 3.96 | 1.08 |
| 43 | s/he works in illegal economic activity | 3.92 | 1.26 |
| 37 | s/he is living in the same place as s/he works with no control over the temperature | 3.80 | 1.26 |
| 72 | s/he is deprived of freely discussing his/her working conditions | 3.68 | 1.18 |
| 29 | his/her contact with other workplaces is restricted | 3.48 | 1.12 |
| | **FINANCE AND MIGRATION** | **4.15** | |
| | **Deductions and migrant work** | **4.40** | **0.28** |
| 50 | his/her wages are withheld | 4.80 | 0.41 |
| 11 | his/her employer charges exorbitant fees for shelter | 4.44 | 0.58 |
| 54 | his/her wages are subjected to illegal deductions | 4.40 | 0.71 |
| 42 | s/he must pay for the right to work | 4.36 | 0.86 |
| 9 | his/her working permit is linked to the current employer | 4.00 | 1.29 |
| | **Misled** | **4.04** | **0.17** |
| 32 | s/he is lied to about his/her rights as a migrant in the country where s/he is working | 4.24 | 0.60 |
| 12 | s/he has had to pay large recruitment fees | 4.08 | 0.81 |
| 41 | s/he has been misled about the type of work | 4.00 | 0.82 |
| 40 | s/he has been misled about the pay | 3.84 | 0.90 |
| | **Wage issues** | **4.00** | **0.35** |
| 16 | s/he receives wages that are insufficient to cover basic needs | 4.40 | 0.87 |

*(Continued)*

**Table 2.** (Continued)

| ID | KEY DIMENSIONS | Rating | |
|----|----------------|--------|--|
| | **Subdimensions and statement labels** | **Mean** | **SD** |
| 8 | s/he does not receive the agreed-upon salary | 4.36 | 0.70 |
| 31 | s/he is lied to about his/her rights as a worker in the place where s/he is working | 4.28 | 0.79 |
| 88 | s/he is not paid equivalent to the minimum wage for his/her work | 4.16 | 0.75 |
| 61 | s/he is not paid regularly and on time | 3.88 | 0.88 |
| 20 | s/he is underpaid for his/her work | 3.76 | 0.93 |
| 57 | s/he is paid below the wage of national workers for the same job | 3.68 | 0.85 |
| 55 | s/he receives below-market wages | 3.48 | 0.82 |
| **HEALTH & SAFETY** | | **3.96** | |
| | **Health, safety and psychosocial hazards** | **3.96** | **0.46** |
| 14 | s/he has no weekly rest from work | 4.56 | 0.71 |
| 74 | s/he experiences verbal abuse | 4.42 | 0.88 |
| 81 | s/he faces humiliation at work | 4.25 | 1.07 |
| 69 | s/he works in unsafe conditions | 4.16 | 0.80 |
| 59 | s/he has to do compulsory overtime | 4.12 | 0.88 |
| 19 | s/he consistently works overtime with no compensation | 4.08 | 0.95 |
| 60 | s/he works in unhealthy conditions | 4.08 | 0.95 |
| 24 | s/he can be harassed | 3.92 | 1.14 |
| 6 | his/her work contract is not renewed unless s/he works extra hours unpaid | 3.88 | 0.88 |
| 51 | s/he has no access to protective equipment | 3.80 | 1.04 |
| 52 | s/he is not trained to use protective equipment correctly | 3.24 | 1.27 |
| 38 | s/he is required to work without proper training | 2.96 | 1.10 |
| **SOCIAL AND LEGAL PROTECTION** | | **3.71** | |
| | **Time-off and legality issues** | **3.93** | **0.32** |
| 2 | s/he has no right to days off | 4.48 | 0.92 |
| 89 | s/he has to work longer hours than the legal maximum | 4.24 | 0.66 |
| 3 | s/he is treated worse than the legally acceptable minimum in the country where s/he works | 4.24 | 0.88 |
| 96 | his/her working conditions do not comply with appropriate national and international legislation | 4.00 | 1.00 |
| 4 | s/he is not granted sick leave | 4.00 | 1.12 |
| 25 | s/he can be discriminated against | 3.80 | 0.91 |
| 7 | s/he has no proper accident insurance covering all possible accidents at work | 3.80 | 1.04 |
| 56 | s/he does not enjoy the rights granted by collectively agreed terms and conditions of employment | 3.68 | 0.90 |
| 23 | s/he may face lower observance of their rights at work | 3.56 | 0.92 |
| 5 | s/he is not granted care leave | 3.52 | 1.26 |
| | **Contract and workload** | **3.74** | **0.51** |
| 93 | s/he suffers labour rights abuse | 4.32 | 0.85 |
| 15 | s/he has no breaks in the daily work routine | 4.04 | 1.06 |
| 1 | s/he does not have a contract with the employer to establish decent wages, hours and working conditions | 3.96 | 0.93 |
| 53 | s/he works an excessive number of hours | 3.88 | 0.83 |
| 63 | s/he does not have a written employment contract | 3.64 | 1.19 |
| 65 | s/he can be dismissed at will | 3.60 | 1.35 |
| 48 | s/he works under pressure | 2.72 | 1.37 |
| | **Health and social benefits** | **3.73** | **0.09** |
| 68 | s/he does not benefit from health coverage | 3.88 | 1.05 |
| 27 | s/he does not have access to health benefits | 3.72 | 0.98 |

(*Continued*)

**Table 2.** (Continued)

| ID | KEY DIMENSIONS | Rating | |
|----|----------------|--------|--|
| | **Subdimensions and statement labels** | **Mean** | **SD** |
| 66 | s/he does not benefit from social protection benefits | 3.72 | 1.14 |
| 86 | s/he has fewer recognised benefits than national workers doing the same job | 3.68 | 0.90 |
| 80 | s/he does not have access to basic social benefits | 3.64 | 1.04 |
| | **Lack of means to get support** | **3.65** | **0.36** |
| 17 | s/he has no right to compensation for injuries and accidents resulting from his/her work | 4.32 | 0.90 |
| 64 | s/he does not have access to formal complaints or dispute resolution procedures | 3.84 | 0.80 |
| 47 | s/he has no capacity to protest or join others in doing so | 3.64 | 0.86 |
| 21 | s/he lacks representation for problems at work | 3.58 | 0.93 |
| 44 | s/he has no ability to engage with a trade union to receive support with legislation issues | 3.56 | 1.04 |
| 30 | his/her contact with migrant associations is restricted | 3.48 | 1.05 |
| 22 | s/he lacks sources of support for problems at work | 3.16 | 1.18 |
| | **Lack of standards enforcement, benefits or information** | **3.51** | **0.45** |
| 70 | s/he is deprived of basic work-related benefits | 3.88 | 0.88 |
| 92 | s/he is denied the main international/national labour standards | 3.88 | 1.01 |
| 26 | s/he does not have access to paid sick leave | 3.84 | 1.03 |
| 85 | s/he has fewer recognized rights than national workers doing the same job | 3.80 | 0.91 |
| 58 | s/he does not benefit from paid leave | 3.68 | 0.99 |
| 33 | s/he is not informed about his/her rights as a worker in the place where s/he is working | 3.56 | 0.96 |
| 90 | s/he does not understand his/her terms of employment | 3.56 | 1.23 |
| 34 | s/he is not informed about his/her rights as a migrant in the country where s/he is working | 3.40 | 1.00 |
| 62 | s/he does not receive a written pay slip detailing pay and deductions | 3.36 | 1.04 |
| 67 | s/he does not benefit from public holidays | 3.32 | 1.18 |
| 91 | s/he has no possibility to make progress in his/her career | 2.32 | 1.07 |

Considering the severity gradient for each of the dimensions with how different aspects of workers' health and wellbeing may be affected by labour exploitation, each dimension of the concept of exploitation revealed in our work seems to align with Maslow's hierarchy of needs [82]. For severe forms of labour exploitation, migrant workers may have unmet physiological needs, as indicated by experts' statements describing poor access to food and water; but also, unmet needs for security and safety, as indicated by their references to cruel working or living conditions. This is in line with the findings of research on human trafficking and modern slavery [9, 37]. For less severe forms of labour exploitation, corresponding to poor employment conditions, migrant workers' self-esteem may be the most affected by labour exploitation. This accords with research from the Employment Conditions Network in the SDH school, where these less severe situations echo the dimension of "enrichment and lack of alienation" [54; p23] of their concept 'fair employment'. The latter is an operationalisation of the concept of decent work for health research. However, the highest needs of Maslow's hierarchy (i.e. self-actualisation and self-transcendence [83]) seem not to be represented on the map. This might relate to the research focus on low-skilled jobs, or to experts not perceiving these jobs as being able to meet such higher needs.

The gradient of severity in the dimensions and statements ratings provides empirical evidence for the hypothesised continuum of labour exploitation between decent work and modern slavery [22]. On the part of the framework rated as highly important to the concept of exploitation, the 'Shelter and personal security' dimension covers situations of threats to, or attacks on migrant workers' fundamental needs, in particular needs for "personal security and

shelter" [82, 83]. It relates to breaches of human rights or criminal law [22] which might characterise situations of modern slavery, and echoes situations of violence highlighted in the Human Rights school [9, 38]. On the lowest rated part of the framework, the 'Social and legal protection' dimension indicates that less severe situations of exploitation correspond to poor employment conditions. It suggests that the lack of rights or rights enforcement represents aspects of labour exploitation less severe compared to those referring to "cruel" treatment. It echoes research on precarious employment in the Social Determinants of Health School that demonstrated associated negative health impacts [84–86]. Dimensions in the middle of the framework are discussed in both schools of thought with some subdimensions being more addressed by the HR school where the rating is higher and by the SDH school where it is lower. Within the second highly rated dimension 'Finance and migration', the subdimensions 'Deductions and migrant work' and 'Misled' echo more with the notion of deception and coercion, as emphasised in the research by the HR school [9, 38]. The 'Health and safety' dimension resonates with both schools: though the SDH school has been the one predominantly addressing occupational health, this is now increasingly investigated in the HR school [6, 63].

## Contributions

The operational framework developed here offers a common middle ground allowing for collaborative health research on labour exploitation [4] between the traditionally separated HR and SDH schools of thought. It allows for a better understanding of the effects of labour exploitation on migrant workers' health, and lays the foundation for multilateral and multidisciplinary dialogues called for by researchers in this field [7, 87–91]. It provides a frame for the HR school to incorporate *"structural drivers"* [6] in research on extreme forms of labour exploitation. The framework also provides a solid basis for global and public health researchers interested in contributing to improve migrant health research and achieve the Sustainable Development Goals (SDGs).

The conceptual framework addresses the first phase of the SOCEPID's framework [66] and could inform research in social epidemiology. Furthermore, it provides a list of statements that can be operationalised into a standardised measure of labour exploitation. The GCM method is described as a robust method to clarify concepts and develop scales likely to be valid and reliable [75]. Each dimension could potentially constitute a section of a questionnaire, the statements items, and their associated ratings a potential score. Further research is needed to test the framework and develop a measure of migrant labour exploitation.

The current research was designed within a pragmatic epistemology that aimed to facilitate the understanding of labour exploitation in the labour market of the 'destination' or 'host' country, with the objective being to measure and identify labour exploitation where the international migrants work. We acknowledge that international migrant workers' exploitation may occur differently at different stages of the migration cycle [92], and our framework may not account for situation in the pre- or post-destination migration phases.

This study therefore lays the foundations of standardised tools to build further evidence in migrant occupational health, such as a monitoring tool for a future multi-stakeholder observatory of migrant health [11].

## Strengths and limitations

This research addresses a conceptual and operational gap in health research on migrant workers' exploitation. It is the first multidisciplinary and participatory conceptual framework of migrant labour exploitation which offers an operationalisable and standardisable framework. It is also the first time that the GCM method, which is a robust methodology [76], has been

used in research in the field of labour exploitation. The use of online data collection allowed for engaging experts from different parts of the world.

The non-random sampling method used may limit the generalisability of the findings as participating experts may have similar political leanings towards exploitation and were mainly from high-income countries. While we identified experts from various regions, we were not able to recruit experts from every region of the world. We did however recruit experts with a broad and international expertise on labour exploitation. We believe that the statements generated would be widely applicable. The framework's novelty is that it offers a tool to capture labour exploitation at the destination country in a holistic and multidimensional way, with items connected to structural aspects of exploitation (e.g. the social and legal dimension) and the aspects connected to management of workers (e.g. exposure to violence). Furthermore, due to the exploratory nature of multivariate analyses, a different sample of experts might modify the dimensions identified. However, the number of participants and stress value obtained compared favourably to other GCM studies [76, 80]. Further research is needed to test the operational framework and assess its validity, reliability and reproducibility.

## Conclusions

Labour exploitation is a multidimensional concept composed of four main dimensions distributed along a continuum of severity. The use of the continuum approach offers new ways of understanding the health needs of migrant workers exposed to labour exploitation. The operational framework described in this paper is standardisable and can foster our understanding of the relationships between exposure to labour exploitation and migrant workers' health. It offers a robust foundation for advancing quantitative research on exploited workers' health, which in turn could inform interventions and policies aiming at improving migrant health and addressing the Sustainable Development Goals (SDGs).

## Supporting information

**S1 File. Dataset for the multivariate and cluster analyses.** The numbers in columns and rows correspond to the number of statements' ID numbers as identified in Table 2 and mapped on Fig 2.
(XLS)

## Acknowledgments

We would like to thank Kuba Misiorny for developing the online platform, as well as Cathy Zimmerman and Ligia Kiss for their initial contributions to this research.

## Author Contributions

**Conceptualization:** Sabah Boufkhed.

**Data curation:** Sabah Boufkhed.

**Formal analysis:** Sabah Boufkhed.

**Funding acquisition:** Sabah Boufkhed.

**Investigation:** Sabah Boufkhed.

**Methodology:** Sabah Boufkhed, Nicki Thorogood, Cono Ariti, Mary Alison Durand.

**Project administration:** Sabah Boufkhed.

**Software:** Sabah Boufkhed.

**Supervision:** Nicki Thorogood, Cono Ariti, Mary Alison Durand.

**Validation:** Sabah Boufkhed.

**Visualization:** Sabah Boufkhed.

**Writing – original draft:** Sabah Boufkhed.

**Writing – review & editing:** Sabah Boufkhed, Nicki Thorogood, Cono Ariti, Mary Alison Durand.

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
