## [Decision Letter · Decision Letter 0]

18 Mar 2022

PONE-D-21-11987

Building a better understanding of labour exploitation’s impact on migrant health: An operational framework

PLOS ONE

Dear Sabah Boufkhed,

Thank you for submitting your manuscript to PLOS ONE. After careful consideration, we feel that it has merit but does not fully meet PLOS ONE’s publication criteria as it currently stands. Therefore, we invite you to submit a revised version of the manuscript that addresses the points raised during the review process.

Please address the following concerns which reviewers raised in your revised manuscript:

Reviewer # 1 asks whether the framework applies to international labor migrants only - non inclusive of internal migrant workers in exploitative conditions. Is this correct?

Further, does the proposed framework of Labour exploitation consider the phases of migrant cycle? The model needs to better consider/integrate/enshrine exploitation not just at the point of destination but also at point of origin prior to departure. Exploitation occurs at country of origin across migration (mis)information, recruitment phases, mandatory health examinations as pre-requisite for labour migration, etc. The exploitation also occurs post-arrival phases for some migrant workers and manifests in forms such as non provision of social welfare support, health and legal protections upon return.

As authors indicate non-random sampling method used may limit the generalizability of the findings as participating experts may have similar political leanings towards the topic and were mainly from high-income countries - with half of the sample being from the UK. Regions such as the Gulf Cooperation Council (GCC) which is one of the main destination regions globally for international migrant workers, and where proportion of migrant to local workers is amongst the highest in the world. Unclear as to the representation from this region.

The second reviewer raises the following (minor) issues. Please address these and revise your manuscript accordingly:

Lines 218-220 state that a second concept map was produced by migrant workers. It was not clear to me if this was included in the results or if it will be analyzed/published separately, but I would like to see more discussion of this in the paper and would be interested to know how it complemented/differed from the other concept map.

The framework proposed by Cwikel (line 170) needs a reference

Line 230: there is a possible grammatical error, as the line does not make sense to reviewer #2.

We look forward to receiving your revised manuscript.

Kind regards,

M. Harvey Brenner, PhD

Academic Editor

PLOS ONE

Journal Requirements:

4. Your abstract cannot contain citations. Please only include citations in the body text of the manuscript, and ensure that they remain in ascending numerical order on first mention.

6. Thank you for submitting the above manuscript to PLOS ONE. During our internal evaluation of the manuscript, we found significant text overlap between your submission and the following previously published works, some of which you are an author.

- https://researchonline.lshtm.ac.uk/id/eprint/4656024/1/2019_PHP_PhD_Boufkhed_S-Copy.pdf

Please revise the manuscript to rephrase the duplicated text, cite your sources, and provide details as to how the current manuscript advances on previous work. Please note that further consideration is dependent on the submission of a manuscript that addresses these concerns about the overlap in text with published work.

We will carefully review your manuscript upon resubmission, so please ensure that your revision is thorough. Please pay special attention to the sections of the results that have overlap.

Reviewers' comments:

Reviewer's Responses to Questions

**Comments to the Author**

1. Is the manuscript technically sound, and do the data support the conclusions?

Reviewer #1: Yes

Reviewer #2: Yes

2. Has the statistical analysis been performed appropriately and rigorously? 

Reviewer #1: Yes

Reviewer #2: N/A

3. Have the authors made all data underlying the findings in their manuscript fully available?

Reviewer #1: Yes

Reviewer #2: Yes

4. Is the manuscript presented in an intelligible fashion and written in standard English?

Reviewer #1: Yes

Reviewer #2: Yes

5. Review Comments to the Author

Reviewer #1: I thought this was a fantastic study and the author was able to present and discuss the results and the complex underlying theories in a pragmatic and clear way. I believe this study to satisfy all the requirements needed to be published in PLOS ONE. The study will be very useful for researchers within the field of labour migration and beyond. A few minor points to consider are listed below:

1) Lines 218-220 state that a second concept map was produced by migrant workers. It was not clear to me if this was included in the results or if it will be analysed/published separately, but I would like to see more discussion of this in the paper and would be interested to know how it complemented/differed from the other concept map.

2) The framework proposed by Cwikel (line 170) needs a reference

3) Line 230: possible grammatical error, the line does not make sense to me

Overall, looking forward to seeing this published!

Reviewer #2: Issues of migrant health and labour exploitation have gained momentum in global political agendas and in the domain of global health AND in the field of migration governance. This is a timely contribution to the area on labour exploitation focusing on migrant workers in manual low-skilled job. Just to confirm, the framework applies to international labour migrants only - non inclusive of internal migrant workers in exploitative conditions - correct?

How does the proposed framework of Labour exploitation consider the phases of migrant cycle? The model needs to better consider/integrate/enshrine exploitation not just at the point of destination but also at point of origin prior to departure. Exploitation occurs at country of origin across migration (mis)information, recruitment phases, mandatory health examinations as pre-requisite for labour migration, etc. The exploitation also occurs post-arrival phases for some migrant workers and manifests in forms such as non provision of social welfare support, health and legal protections upon return.

As authors indicate non-random sampling method used may limit the generalizability of the findings as participating experts may have similar political leanings towards the topic and were mainly from high-income countries - with half of the sample being from the UK. Regions such as the Gulf Cooperation Council (GCC) which is one of the main destination regions globally for international migrant workers, and where proportion of migrant to local workers is amongst the highest in the world. Unclear as to the representation from this region.

6. PLOS authors have the option to publish the peer review history of their article (what does this mean?). If published, this will include your full peer review and any attached files.

Reviewer #1: **Yes: **Isabelle Pearson

Reviewer #2: **Yes: **Dr Kolitha Wickramage

---

## [Author Response · Author response to Decision Letter 0]

26 Apr 2022

Reviewers’ comments followed by the Authors’ reply (starting with ->)

Reviewer #1 

-> Thank you very much for your feedback and request for clarification. This has helped strengthen our manuscript. 

Reviewer # 1 asks whether the framework applies to international labor migrants only - non inclusive of internal migrant workers in exploitative conditions. Is this correct? 

-> Thank you for your feedback. The framework was indeed designed for international labour migrants as a first step. Further research would be needed to assess whether the framework could be used in internal migration context. 

We clarified this in the core text after the definition we use for ‘migrant workers’, as follows 

“This paper will focus on international migrant workers.” (line 56)

Further, does the proposed framework of Labour exploitation consider the phases of migrant cycle? The model needs to better consider/integrate/enshrine exploitation not just at the point of destination but also at point of origin prior to departure. Exploitation occurs at country of origin across migration (mis)information, recruitment phases, mandatory health examinations as pre-requisite for labour migration, etc. The exploitation also occurs post-arrival phases for some migrant workers and manifests in forms such as non provision of social welfare support, health and legal protections upon return. 

-> Thank you for your comment. The framework has indeed considered the phases of migrant cycle, and we have now clarified this in the discussion as follows:

“The current research was designed within a pragmatic epistemology that aimed to facilitate the understanding of labour exploitation in the labour market of the ‘destination’ or ‘host’ country, with the objective to measure and identify labour exploitation where the international migrants work. We acknowledge that international migrant workers’ exploitation may occur differently at different stages of the migration cycle [92], and our framework may not account for situation in the pre- or post-destination migration phases.” (lines 433-38)

As authors indicate non-random sampling method used may limit the generalizability of the findings as participating experts may have similar political leanings towards the topic and were mainly from high-income countries - with half of the sample being from the UK. Regions such as the GCC which is one of the main destination regions globally for international migrant workers, and where proportion of migrant to local workers is amongst the highest in the world. Unclear as to the representation from this region.

-> Thank you for this comment. 

During the identification of experts, we mapped out experts from and of every region of the world. Unfortunately, we did not have responses from all regions as mentioned in the discussion. 

While we were not able to recruit experts from every region of the world, we did however recruit experts with broad and international expertise on labour exploitation. We believe that the statements generated would be widely applicable. The framework’s novelty is that it offers a tool to capture labour exploitation at the destination country in a holistic and multidimensional way, with items connected to structural aspects of exploitation (e.g. the social and legal dimension) and the aspects connected to management of workers (e.g. exposure to violence).

For example, the Kafala system that is still implemented in several countries in the Middle East and GCC would result in situations captured in several dimensions of labour exploitation described in the framework, such as 

- the lack of Social and Legal Protection (e.g. “s/he can be dismissed at will”; s/he does not benefit from social protection benefits; “s/he is treated worse than the legally acceptable minimum in the country where s/he works”) 

- Health and Safety issues (e.g. “s/he works in unsafe conditions”)

- Issues related to Finance and Migration (e.g. “his/her working permit is linked to the current employer”; s/he is lied to about his/her rights as a migrant in the country where s/he is working; s/he is paid below the wage of national workers for the same job)

- Issues related to Shelter and Personal security (e.g. s/he is threatened with deportation”; “s/he is obliged to work under cruel or inhumane conditions”)

The framework’s novelty is that it offers a tool to capture labour exploitation at the destination country in a holistic and multidimensional way, with items connected to structural aspects of exploitation (e.g. the social and legal dimension) and the aspects connected to management of workers (e.g. exposure to violence). The framework provides a much-needed basis for future research that could test whether it is applicable to GCC countries. 

We added the following specification in the discussion: 

“While we identified experts from various regions, we were not able to recruit experts from every region of the world. We did however recruit experts with a broad and international expertise on labour exploitation. We believe that the statements generated would be widely applicable. The framework’s novelty is that it offers a tool to capture labour exploitation at the destination country in a holistic and multidimensional way, with items connected to structural aspects of exploitation (e.g. the social and legal dimension) and the aspects connected to management of workers (e.g. exposure to violence).” (lines 451-57)

- Reviewer #2 

The second reviewer raises the following (minor) issues. Please address these and revise your manuscript accordingly:

 -> Thank you very much for the positive feedback and supportive comments.

Lines 218-220 state that a second concept map was produced by migrant workers. It was not clear to me if this was included in the results or if it will be analyzed/published separately, but I would like to see more discussion of this in the paper and would be interested to know how it complemented/differed from the other concept map.

 -> Thank you very much for highlighting this point. We indeed performed a Concept Mapping (GCM) with migrant workers. The GCM method was adapted for the population that is considered vulnerable and hard-to-reach, and was conducted in person. Given the word limitations and the difference in method, we could not include it as part of one paper. It will constitute a stand-alone paper that would bring the voices of migrant workers. In order to avoid confusion for the reader, we adapted the lines mentioned by reviewer #2 as follows: 

“Migrant workers are also considered experts on the topic [74], and we have undertaken further research with international migrants working in low skilled jobs in London to assess their conceptualisation [19].” (lines 209-11)

The framework proposed by Cwikel (line 170) needs a reference

 -> Thank you very much for pointing this out. 

We have now amended the references. 

Line 230: there is a possible grammatical error, as the line does not make sense to reviewer #2. 

-> Thank you very much for pointing this out, there are indeed few words missing. We amended as follows: 

“It is expected that the statements generated for the sorting-rating exercise are varied enough to cover all aspects of the concept to be mapped, and it is common to include new participants who did not participate in the first phase (3,4).” (lines 221-22)

---

## [Decision Letter · Decision Letter 1]

11 Jul 2022

Building a better understanding of labour exploitation’s impact on migrant health: An operational framework

PONE-D-21-11987R1

Dear Dr. Boufkhed,

We’re pleased to inform you that your manuscript has been judged scientifically suitable for publication and will be formally accepted for publication once it meets all outstanding technical requirements.

Kind regards,

M. Harvey Brenner, PhD

Academic Editor

PLOS ONE

Additional Editor Comments (optional):

Reviewers' comments:

Reviewer's Responses to Questions

**Comments to the Author**

1. If the authors have adequately addressed your comments raised in a previous round of review and you feel that this manuscript is now acceptable for publication, you may indicate that here to bypass the “Comments to the Author” section, enter your conflict of interest statement in the “Confidential to Editor” section, and submit your "Accept" recommendation.

Reviewer #1: All comments have been addressed

Reviewer #2: All comments have been addressed

2. Is the manuscript technically sound, and do the data support the conclusions?

Reviewer #1: (No Response)

Reviewer #2: Yes

3. Has the statistical analysis been performed appropriately and rigorously? 

Reviewer #1: (No Response)

Reviewer #2: N/A

4. Have the authors made all data underlying the findings in their manuscript fully available?

Reviewer #1: (No Response)

Reviewer #2: Yes

5. Is the manuscript presented in an intelligible fashion and written in standard English?

Reviewer #1: (No Response)

Reviewer #2: Yes

6. Review Comments to the Author

Reviewer #1: (No Response)

Reviewer #2: All review feedback incorporated in revised version. I am happy for this to proceed for publication, thank you.

7. PLOS authors have the option to publish the peer review history of their article (what does this mean?). If published, this will include your full peer review and any attached files.

Reviewer #1: **Yes: **Isabelle Pearson

Reviewer #2: **Yes: **Dr Kolitha Wickramage

---

## [Editor Report · Acceptance letter]

22 Jul 2022

PONE-D-21-11987R1 

Building a better understanding of labour exploitation's impact on migrant health: An operational framework 

Dear Dr. Boufkhed:

I'm pleased to inform you that your manuscript has been deemed suitable for publication in PLOS ONE. Congratulations! Your manuscript is now with our production department. 

Kind regards, 

on behalf of

Professor M. Harvey Brenner 

Academic Editor

PLOS ONE